# Double-Layer Micro Porous Media Burner from Lean to Rich Fuel Mixture: Analysis of Entropy Generation and Exergy Efficiency

**DOI:** 10.3390/e23121663

**Published:** 2021-12-10

**Authors:** Nazmi Che Ismail, Mohd Zulkifly Abdullah, Khairil Faizi Mustafa, Nurul Musfirah Mazlan, Prem Gunnasegaran, Agustinus Purna Irawan

**Affiliations:** 1School of Mechanical Engineering, Engineering Campus, Universiti Sains Malaysia, Nibong Tebal 14300, Malaysia; nazmi@student.usm.my (N.C.I.); mekhairil@usm.my (K.F.M.); 2School of Aerospace Engineering, Engineering Campus, Universiti Sains Malaysia, Nibong Tebal 14300, Malaysia; nmusfirah@usm.my; 3Institute of Power Engineering, Putrajaya Campus, Universiti Tenaga Nasional, Jalan IKRAM-UNITEN, Kajang 43000, Malaysia; Prem@uniten.edu.my; 4Department of Mechanical Engineering, Universitas Tarumanagara, Jl. Letjen S. Parman No.1, Jakarta Barat 11440, Indonesia; agustinus@untar.ac.id

**Keywords:** thermal efficiency, exergy efficiency, entropy generation, submerged flame, lean to rich mixtures, premixed butane-air combustion

## Abstract

Porous media burner (PMB) is widely used in a variety of practical systems, including heat exchangers, gas propulsion, reactors, and radiant burner combustion. However, thorough evaluations of the performance of the PMB based on the usefulness of entropy generation, thermal and exergy efficiency aspects are still lacking. In this work, the concept of a double-layer micro PMB with a 23 mm cylindrical shape burner was experimentally demonstrated. The PMB was constructed based on the utilization of premixed butane-air combustion which consists of an alumina and porcelain foam. The tests were designed to cover lean to rich combustion with equivalence ratios ranging from ϕ = 0.6 to ϕ = 1.2. It was found that the maximum thermal and exergy efficiency was obtained at ϕ = 1.2 while the lowest thermal and exergy efficiency was found at ϕ = 0.8. Furthermore, the findings also indicated that the total entropy generation, energy loss, and exergy destroyed yield the lowest values at ϕ = 1.0 with 0.0048 W/K, 98.084 W, and 1.456 W, respectively. These values can be stated to be the suitable operating conditions of the PMB. The findings provided useful information on the design and operation in a double-layer PMB.

## 1. Introduction

Combustion technology is quickly evolving as a result of the increased demand in our daily life and therefore posing a danger to the depletion of fossil fuels. One of the most common uses of fossil fuels is in home burners and industrial radiant burners. This issue has impacted the demand for fossil fuels and has markedly increased the price, which cannot be tolerated. Moreover, these fuels commonly generate the emission pollutions such as carbon monoxide (CO) and nitrogen oxide (NO_x_) that negatively affect the environment and create health problems [1,2]. Thus, one of the methods that was suggested by researchers was to use porous media burners (PMBs) and it has proven to be an ideal solution to minimize the above-stated problems. The household burner has piqued the curiosity of many academics and it has been given a top priority in the last two decades [3,4,5]. As a result, the research concept forwarded here focuses on the investigation and analyses of a double-layer micro PMB technology that is suited for premixed butane-air combustion. 

Due to their high heat capacity, conductivity, and emissivity, porous media burners (PMB) have gained popularity and are frequently utilized in household products. It has been discovered that a burner with the incorporation of porous media gives better power density and modulation, as well as an increase in the combustion efficiency, lower pollution, and higher heat transfer rate [6,7,8,9]. When the porous media is inserted into a free flame burner, it promotes greater heat transmission with stable flame conditions and results in a higher flame temperature compared with a conventional burner without a porous media [2]. 

A laminar premixed flame is a low-velocity combination of a fuel and an oxidant mixture before entering the combustion zone. Flame velocity, *S_L_*, is one of the most important factors in determining the laminar premixed flames. If the outlet flow velocity at the burner is low, the flow of the mixture in the burner will be laminar [10]. The resulting flame velocity will be a laminar flame velocity. The propagation of free-flame velocity at steady state, unstretch flame, parallel flows, and adiabatic flame temperature which propagates relative to the premixed mixture is characterized as a laminar flame velocity [11]. In porous medium combustion, laminar flame velocity is used to explain the combustion phenomena such as flame stability, flashback, and flame blowout.

Bunsen burners [12,13] and flat flame burners [14,15] are the two types of stationary burner methods used to determine the laminar premixed flame combustion. These studies focused on the flat flame burner that used porous media foams. The flat flame burner employs porous media materials as a diffuser by placing the porous media on the top of the burner rim, resulting in a constant profile of the gas flow rate which creates a flat flame [16]. This approach was employed at low gas flow rates and allowed for the measurement of small values of laminar flame velocities that occur in combustion mixtures near the flammability limit.

Various investigations on the characteristics of porous media combustion have been conducted in the last several decades. The investigations are primarily related to enhanced combustion efficiency in dealing with different types of porous media materials, porosity, and burner size. Each of these characteristics has an essential role to play in improving the burner efficiency of the household stove. Yu et al. [17] studied the combustion properties and thermal efficiency of three types of porous media in order to study the performance and emissions of a rectangular shape combustor burner. Due to the variations in the porosity, the metal fiber (MF) burner was found to influence considerably on the thermal efficiency, which was approximated to be 89 %. Moreover, metal fiber (MF) gives the lowest CO emission compared to ceramic (CM) and stainless steel fin (SF). Wang et al. [18] examined the effect of the temperature variation on the premixed methane-air combustion using alumina pallets with varying diameters. They discovered that an increase in the diameter of the pellets resulted in a reduction in peak temperature at the reaction zone but increase the temperature in the preheating zone due to the differences in the porosity. In addition, Su et al. [19] examined the heat recovery rate of hydrogen combustion using various types of porous media such as oxide-bonded silicon carbide, aluminum oxide, and zirconia with 60 and 30 pores per inch (PPI). They discovered that the oxide-bonded silicon carbide with a low equivalence ratio has the best heat recovery and reaction temperatures. There are also a few research works on the efficiency of PMB in different burner sizes. Nozari et al. [20] examined the flame stability, efficiency, and NO_x_ emissions in premixed combustion with burner diameters of 5.08 cm and 7.62 cm using an inert silicon-carbide (SiC) porous block. They discovered that a smaller burner diameter resulted in a longer flame length, a narrower flame stability region, and higher power output density.

Many scholars have examined the efficiency and emission in relation to the equivalence ratio, ϕ in order to improve the performance of PMB based on the efficiency and emission. The equivalence ratio is a key parameter in determining whether the combustion is lean or rich within the PMB. One of the benefits of utilizing PMB is that it can be burned in lean mixtures. The research works on PMB systems in both lean and rich mixtures employing a double-layer porous medium are given in Table 1.

The primary features in assessing PMB performance and minimizing irreversibility are the assessment of entropy generation and exergy efficiency. These characteristics use the energy in combustion and cause an exergy loss to the environment, which cannot be overlooked [28,29]. According to Jiang et al. [30], the entropy production rate in PMB systems must be minimized in order to limit the irreversible process and exergy loss in porous medium combustion systems. The irreversible processes include fluid friction, heat conduction, chemical reaction, and mass dissipation. Jejurkar and Mishra [31,32] investigated the influence of entropy production rate on the wall thermal conduction using a numerical analysis and concluded that the chemical reaction and mass diffusion were the primary effects in the entropy formation in H_2_-air premixed flame combustion. On the other hand, Acampora and Marra [33] discovered that the chemical reactions were not necessarily the primary contributors to the overall entropy production, but the influence of pressure increment plays an essential role in determining the chemical reactions and heat conduction in the combustion. Furthermore, Wang et al. [34] discovered that the chemical reactions contribute significantly to the overall entropy production, followed by heat conduction and mass diffusion. They claimed that the overall entropy production rate of premixed fuel-air combustion increase with an increasing in the intake flow velocity. Other research work by Ni et al. [35] discovered that the chemical reactions provide 70% of the overall entropy production, whereas heat conduction contributes only 15% in micro thermo-photovoltaic (MTPV) systems. According to Mohammadi and Ajam [36], heat transfer conduction affected the most to entropy formation, followed by chemical processes, mass diffusions, and fluid friction.

To improve the burner performance, the entropy generation rate in the combustion systems must be reduced to enhance thermal and exergy efficiency. In reality, only a few scholars have attempted to increase the performance of domestic burners based on the second law of efficiency related to the porous media combustion [37]. It was discovered that the household burner with porous medium produced higher thermal efficiency, with an average of 10% more than the standard ordinary household burners available in the market. The exergy analysis has recently been adopted in PMB combustion to improve heat management. Rana et al. [38] used a cylindrical micro combustor to numerically evaluate the exergy analysis and flame speed of premixed methane-air combustion in heat recirculation. They discovered that the second law efficiency offered valuable tools in predicting combustion performance. Nadimi et al. [39] examined the impact of inserting a micro-fin within the micro-combustor to increase the exergy efficiency of the combustion, whereas Li et al. [40] focused on inserting a rectangular rib in the combustion chamber to increase the exergy efficiency of the micro-combustor. According to Huang et al. [41], the exergy efficiency of the premixed propane/hydrogen flame dropped as channel height increase in the micro-planar combustor. Cai et al. [42] recently revealed that the exergy efficiency using an inserted bluff-body within a combustor is found to be greater than the conventional combustor. As a result, the current study might serve as a guideline for future research into the development of porous media combustion burner systems from the standpoint of the second law of thermodynamics. 

The current work aims to investigate the entropy generation, thermal and exergy efficiency in a double- layer PMB that utilizes premixed butane-air combustion, with a detailed equivalence ratio, ϕ. The influence of the change in equivalence ratios from lean mixtures (ϕ = 0.6) to rich mixtures (ϕ = 1.2) in flame characteristics, temperature variation, emissions, energy loss, and exergy destroyed are the primary responsible variables that affected the performance of the PMB combustion. The results of this research provided useful information on the design and operation of PMBs, as well as substantial knowledge of the combustion phenomena in a double-layer PMB.

## 2. Methodology

### 2.1. Experimental Setup and Materials

Figure 1 shows a schematic design of the premixed butane-air PMB utilized in this study. The experimental design utilized in this study was the same as that used in earlier study by Janvekar et al. [43]. The burner (7) was made of mild steel and it is cylindrical in shape, measuring 100 mm in length, 23 mm in internal diameter (ID), and 2 mm in thickness. In this research, two types of porous media materials (10), an alumina foam and a porcelain foam, were developed and tested. The alumina foam was produced by Goodfellow Cambridge Limited (LS 3699006/1), UK and it has eight pores per centimeter (ppcm), with 84 percent porosity, and a thickness of 15 mm. In contrast, a porcelain foam was produced by the School of Material and Mineral Resources Engineering, Universiti Sains Malaysia, with 26 pores per centimeter (ppcm), 86 percent porosity, and a thickness of 10 mm. The double-layer PMB features porous media materials on top of the burner, with an alumina foam acts as the reaction zone (top layer) and the porcelain foam acts as the preheat zone (bottom layer) as same method configurations align with several published works [3,19,37]. Both porous media materials were chosen because they have high heat resistance and they can be easily shaped into variety of forms to fit a cylindrical shape burner.

The mixing unit (5) at the bottom of the PMB functions as a premix region for both butane fuel and air prior to the entrance of the combustion burner. The mixing unit was used to improve the mixing enhancement between the butane fuel and air by employing wire steel mesh. The experiments were carried out using a butane cartridge (2) which was manufactured by Milux Sdn. Bhd. (Malaysia) as a fuel source. The natural air was provided using a commercial aquarium air pump (1) which was manufactured by Atman Co., Ltd. (China). Both the butane fuel and air were monitored using the mass flow meters (3) and a Vögtlin Instruments (Switzerland) was used as a control valve to vary the equivalence ratio in both lean and rich combustion regions. In addition, the butane fuel and air were delivered through a rubber tube that was connected to the combustion burner continuously.

A Fluke Ti27 (11) industrial commercial thermal imager (Netherlands) and a Nikon Corp. digital camera (Japan) were used to capture the thermal images and photographic pictures, respectively with equivalence ratios ranging from 0.6 to 1.2. The emissions produced by the PMB burner were measured using a Kane 251 (6) portable flue gas analyzer (UK) to record the amount of carbon monoxide (CO) and nitric oxide (NO) in parts per million (ppm).

Nine K-type thermocouples (9) were used to measure the temperatures, with four thermocouples (at different heights, Z of 1 mm, 9 mm, 17 mm, and 31 mm) were placed on top of the PMB for measuring the flame temperature. Five thermocouples (labelled as T1, T2, T3, T4, and T5) were used to measure the porous wall temperature for both the alumina and porcelain foam, as shown in Figure 2. All thermocouples used in this experiment were linked using an Advantech USB-4718 data collecting device (4), and the temperature data was saved to a Hewlett-Packard Pavilion p6345d personal computer (12).

Initially, the premixed butane-air mixture was ignited on top of the burner using external ignition electrodes. After ignition, the butane fuel and air were adjusted to the correct flow rates using a digital flow meter to provide the exact values of equivalence ratio. The experimental value of the equivalence ratio was increased by 0.1 from lean to rich mixtures, covering a range of equivalence ratios from 0.6 to 1.2. To maintain steady-state conditions, the premixed butane-air data collection and an exhaust gas monitoring were performed after 30 min for each experiment. This method prevented the blow-off of the burner or flashback throughout the combustion process same method made by previous researchers [43]. Porous media materials were fixed to the alumina foam as the reaction zone and to the porcelain foam as the preheat zone, using a 23 mm diameter burner size and a double-layer configuration. The equivalence ratio was varied in each experiment, which was carried out at room temperature, in order to achieve the desired PMB characteristics and performance.

### 2.2. Parameter Studies

Table 2 summarizes the pertinent parameters which were covered in this investigation on a 23 mm double-layer PMB with equivalence ratios ranging from lean to rich combustion. In the experiments, the equivalence ratio was determined from the mass flow rate of both the butane fuel and air supplied. As a result, the detailed equivalence ratio ranged from ϕ = 0.6 (lean mixtures) to ϕ = 1.2. (rich mixtures). The laminar flame velocity, *S_L_*, given in Equation (1), was determined from each equivalence ratio based on the volumetric flow rate of the fuel mixture and the cross-sectional area of the PMB [44]. Furthermore, if the laminar flame velocity is known, the Peclet number, *Pe*, and Reynolds number, *Re* can be calculated, as shown in Equations (2) and (3).
(1)$$S_{L}=\frac{{ύ}_{C_{4}H_{10}}+{ύ}_{air}}{A}$$
where ${ύ}_{C_{4}H_{10}}$ and ${ύ}_{air}$ denote the volumetric flow rates of butane fuel and air, respectively, and A denotes the inner cross-sectional area of the burner housing.

The flame stabilization and propagation in the porous media and the installation of the porous media inside the burner housing containing the porous media foams, were categorized based on the *Pe* number. The *Pe* number was utilized to enable the combustion to operate in lean mixtures in order to prevent flashback inside the PMB. For safety purpose and effectiveness of the combustion, *Pe* > 65 for flame propagation was applied in the reaction zone and *Pe* < 65 for flame quenching used in the preheat zone [45]. The Peclet number, *Pe*, was computed using Equation (2) [46]:(2)$$Pe=\frac{S_{L}\;D\;\rho \;C_{p}}{k}$$
where, D denotes the pore diameter of the porous material, ρ is the gas mixture’s density, $C_{p}$ is the specific heat of the butane gas mixture, and k denotes the thermal conductivity coefficient. The Reynolds number during the flow fuel mixture can be expressed by Equation (3):(3)$$Re=\frac{\left(\rho_{C_{4}H_{10}}+\rho_{air}\right)\;S_{L}\;d}{\left(\mu_{C_{4}H_{10}}+\mu_{air}\right)}$$
where d is the inner diameter of the burner housing and μ is the dynamic viscosity of the gas mixture, respectively. According to Table 2, the flow mixture is turbulent because the minimum value for the *Re* number for the fuel mixture is 421, indicating that the flow fuel mixture is more than Re > 100 [23].

### 2.3. Thermal Efficiency Measurement and Second Law of Thermodynamics Data Calculation 

From the combustion reaction, simple assumptions for the energy and exergy principles were applied to calculate the performance efficiency of the PMB. The PMB performance was calculated based on the conservation law of energy using water boiling test, in which energy supplied, Q_total_ should be equal to the energy produced by the combustion, Q_actual_. This approach was the same as that employed in our earlier work by the main authors [47]. The mass flow rate, ṁ, was derived from the density, ρ, and volumetric flow rate, ύ, of butane fuel. Energy supplied, Q_total_ values were derived from the mass flow rate, ṁ, and the calorific value, C_v_, of butane fuel.
ṁ = ρ × ύ(4)
Q_total_ = ṁ × C_v_
(5)

After starting the combustion, the fuel and air flow rates were adjusted to reach the required equivalence ratio, ϕ. The mass of water, M_w_, and the container, M_c_, were determined using a digital weighing scale and then filled with a known mass of water (0.5 kg) at room temperature. The ambient temperature of the water was measured with a mercury thermometer from Alla (France). The container was put on the burner, and after the flame has stabilized, the time, t, was recorded using a digital stopwatch, and the temperature of the water in the container was monitored until it reached the final temperature of 50 °C.
Q_actual_ = [(M_w_C_w_ + M_c_C_p_) (50 °C − T_o_)]/t (6)
where, C_w_ and C_p_ are the standard values of the specific heat of the water and container, C_w_ = 4.1826 kJ/kg.K and C_p_ = 0.5024 kJ/kg.K, respectively. As a result, using Equation (7), the thermal efficiency is the ratio of energy produced divided by total energy:η_thermal_ = (Q_actual_/Q_total_) × 100%(7)

The basic thermodynamic calculations provide direct derivation of the entropy generation and exergy analysis in the combustion process. As a result, the energy loss from the system can be presented as follows:E_loss_ = Q_total_ − Q_actual_(8)

The total exergy in the combustion is then:Q_exergy_ = (Q_actual_/ṁ) − T_o_ (S_max_ − S_amb_)(9)
where, S_max_ is the entropy at maximum temperature, T_max_ while, S_amb_, is the entropy at an ambient temperature, T_o_, respectively. 

In this scenario, the total entropy generation, S_gen_, may be calculated using the same formula as made by [28,48]: S_gen_ = ṁ × Δs (10)
where Δs is the specific entropy different, S_max_ − S_amb_.

The exergy is destroyed and thus E_des_ during the combustion process is proportional to the total entropy generation, and it can be stated as follows:E_des_ = S_gen_ × T_o_
(11)
where T_o_ is the ambient temperature, 302 K. 

As a result, the exergy efficiency is defined as follows: η_exergy_ = (Q_exergy_/Q_total_) × 100%(12)

### 2.4. Analysis of Uncertainty

The random error is predicted to occur in all of the experimental data collected in this study. It is also important to evaluate the accuracy of the measured values. As a result, the uncertainty analysis depends on the random error and it must be analysed using statistical measurements and calculations. All experiments were repeated three times to ensure the accuracy of the collected data, and the averages of these data were calculated to be the part of the error analysis. For all experimental data, the mean value ($\bar{x}$), and the standard deviation ($\sigma_{x}$), are expressed as follows:(13)$$\bar{x}=\frac{1}{n}\overset{n}{\underset{i=1}{\sum }}X_{i}$$
(14)$$\sigma_{x}={\left[\frac{1}{n-1}\overset{n}{\underset{i=1}{\sum }}{\left(X_{i}-\bar{X}\right)}^{2}\right]}^{\frac{1}{2}}$$

Table 3 shows the uncertainty calculation based on the mean ($\bar{x}$), standard deviation ($\sigma_{x}$), and standard error ($\sigma_{\bar{x}}$). The uncertainty (U_n_) was calculated by calculating the percentage ratio of standard deviation divided by mean value. The highest uncertainty was found to be 3.627% at a thermocouple height of 31 mm, which is considered low in the random error analysis.

## 3. Results and Discussions

### 3.1. Flame Stabilization and Thermal Imager

The initial experiments were conducted using a 23 mm double-layer PMB, at different values of equivalence ratios to observe the flame stability. In this study, the mass flow rate for butane-air premixed mixture was adjusted to achieve the desired equivalence ratio. The use of porous media in combustion burners offers the benefits to the PMB by allowing the burner to function in lean mixtures with a minimal quantity of fuel mixture to create optimal performance. Figure 3 shows photographic and thermal pictures of a double-layer PMB at several values of equivalence ratios, captured with a digital camera and a portable thermal imager. Based on visual inspection, the equivalence ratio of 0.6 < ϕ < 0.8, indicates that the double-layer PMB has a submerged flame region, whereas in the range of the equivalence ratio of 0.9 ˂ ϕ ˂ 1.2, indicates that the double-layer PMB has both submerged and surface flame regions. The surface flame appears to emerge at ϕ = 0.9, with a tiny flame on top of the burner rim. Furthermore, the burner housing (mild steel) began to glow in red color as the equivalence ratio start to rise from 0.6 to 0.8, then gradually reduced to the faded red color as the equivalence ratio increase from 0.9 to 1.0. The submerged flame finally disappeared as the equivalence ratio move from 1.1 to 1.2 and surface flame start to developed at ϕ = 0.9. It can be observed that at ϕ = 1.2, the surface flame has completely developed and achieved steady condition, producing a red flame color.

On the other hand, the thermal images display red contour color for the alumina foam (reaction zone) and yellow contour color for the porcelain foam (preheat zone) from ϕ = 0.6 to ϕ = 1.2. It can be observed that the heat loss by radiation and conduction from the solid wall of the PMB to the environment occur during combustion (purple contour color around the burner housing). The radiation heat began to increase as the equivalence ratio is increase from 0.6 to 0.9, and it decrease as the equivalence ratio is increase from 1.0 to 1.2. As the heat recirculated inside the PMB, the temperature distribution began to shift downward from the top burner housing to the bottom part. This implies that the fuel-air mixture that enters the burner housing begins to heat up before igniting inside the porous material. When we compare to the free flame (with no porous added), this observation indicates an improvement in the performance efficiency of the household burners.

The difference in porosity between the reaction and preheat zones is an essential element in controlling the burner stability, which influences the development of submerged flames and the location of surface flames [44]. The submerged flame occurs in the reaction zone as the pores size is bigger (8 ppcm) in diameters. Only at the equivalence ratios of 0.6 and 0.7 (shown in Table 3) match the *Pe* number guidelines for the placement of porous media in the reaction and preheat zones. As a consequence, at ϕ = 0.8 and 0.9, an increase in the porous wall temperature with higher production of CO emissions coupled with an unintended noise can be attributed to the unbalanced in the premixed mixture that might be quenched inside the porous media. This happens because when the equivalence ratio decreases, the burning velocity drops and the air flow rate rises. The higher red heat on PMB can be a problem that will affect the durability of the alumina and porcelain foam with long exposure. In conclusion, the equivalence ratio of the entering fuel-air mixtures influences the flame stability and flame temperature on PMB.

### 3.2. Temperature Distributions of PMB

The temperature distribution of the flame is a key element in influencing the thermal and exergy efficiency of a double-layer PMB. Figure 4 depicts the flame temperature at various thermocouple heights, Z from the burner top surface (as indicated in Figure 2), with the Z-axis orientation changing as the equivalence ratio was gradually changed from lean to rich mixtures. The surface flame temperature is determined by measuring the temperature with thermocouples mounted on the top of the burner rim (as Z: 1 mm, 9 mm, 17 mm, and 31 mm). The flame location was calculated by determining the position of the flame temperature that achieved the highest values among four thermocouples with a variation in the equivalence ratios. The results demonstrate that the flame location did not change from 0.6 to 1.1 at Z: 1 mm equivalence ratio. With an increase in the flame velocity and butane flow rate at ϕ = 1.2, the flame location tended to move upward from Z: 1 mm to Z: 9 mm. 

The surface flame temperature decreases at equivalence ratios of 0.6 < ϕ < 1.0 (lean mixtures) for Z: 1 mm, while for Z: 9 mm, the surface flame temperature is virtually constant in flame temperature, and Z: 17 mm and 31 mm rise in flame temperature. The results revealed that when the equivalence ratio decrease from ϕ = 1.0 to ϕ = 0.6, at Z: 1 mm (top surface of the burner rim), the flame temperature increase. It happens as a result of high temperature double-layer porous media foam which serves as a medium of rapid solid-gas heat recirculation inside the burner. As a result, high heat recirculation caused a rapid increase in the flame temperature of the incoming butane-air mixture, allowing it to self-ignite. In addition, a marginal increase in the butane flow rate leads to corresponding release of greater combustion heat power. Surprisingly, at ϕ = 1.0 (stoichiometric mixture), the flame temperature is nearly constant for all four thermocouples, Z with little fluctuations. The flame temperature has reached the steady state condition and the trend of the temperature profiles supports the fact that the plateau of combustion is attained at the stoichiometric mixture. It also denotes that the flame is steady inside the porous medium as a result of internal heat recirculation combustion of incoming air-fuel mixtures at acceptable circumstances. This is known as flat flame behavior same reported by Giurcan et al. [16], in which the double-layer PMB acts as a flat flame burner by placing the porous media as a diffuser on the top of the burner rim, creating a constant profile of the air-fuel mixtures flow rate. 

The surface flame temperature for all four thermocouples at an equivalence ratio of 1.0 < ϕ < 1.2 (rich mixtures) begins to grow from ϕ = 1.1 to 1.2 owing to an increase in the butane fuel flow rate, which caused a concomitant increase in the heating value of the combustion. It implies a significant increase in the heat release rate from PMB combustion. A primary contributing factor could also be traced to a minute increase in the butane flow rate in the rich regions of combustion. The conduction and convection modes of combustion in the reaction zone are significantly dominant and the largest temperature increase is subsequently more evidenced. The reaction rate could have been impacted because of the slow flame velocity (Table 2) but the effect of the residence time has dominated the combustion performance of the PMB. We argue that the installation of an alumina foam in the reaction zone has somewhat stabilized and the corresponding change in the equivalence ratio toward richer mixture has further enhanced the combustion performance. Heat transfer from the combustion zone via convection was carried by the exhaust gas. At a higher distance from the top of the burner (an increasing value of Z), the convection loss comparatively increased and resulted in a lower temperature recorded for Z = 31 mm. In addition, radiative heat loss from the top of the hot burner exit to downstream environment is also noticeable and leads to the reduction in the recorded temperature as explained above. 

The porous wall temperature is a measurement of the heat generated by the porous media foam in alumina and porcelain from the side of the double-layer PMB, and it is affected by the flame velocity in preheat and reaction zones. Figure 5 depicts the porous wall temperature of different thermocouple locations (from T1 to T5) as a function of the equivalence ratio from lean to rich mixture in a double-layer PMB. The findings reveal a pattern of increasing porous wall temperature with increasing equivalence ratios from 0.6 to 0.9 and then beginning to decrease from ϕ = 0.9 to 1.2, except for the T2 (middle alumina). With increasing equivalence ratios owing to flame stabilization inside the double-layer PMB, the porous wall temperature for all thermocouples seems to be virtually constant (not much temperature variation). The highest porous wall temperature achieved from T3 (bottom alumina) where the flame is easily stabilized between the interface of the two layers of porous foam before the flame moves upward to the top of the burner rim. 

Normally, in double-layer PMB, the reaction zone produces higher porous wall temperature than in preheat zone due to the variations in porosity and pore diameter of the porous media. Increase pore diameter can lead to increase void volume and decrease solid matrix surface area. As a result of pore diameter variations, more heat was transferred from the reaction zone (large pore) to the preheat zone (small pore) via convection and radiation. Furthermore, when the equivalence ratio increases from 0.6 to 1.2, the heat loss as the porous wall temperature increases via the wall burner is more dominating in lean mixtures than rich mixtures. As a consequence, the submerged flame occurs in lean mixtures, with the burner wall turning red (as shown in Figure 3), and the solid matrix temperature rises to a peak at the alumina foam (downstream), leading the submerged flame to stable in double-layer PMB.

### 3.3. The Characteristics of CO and NO Emissions 

The CO and NO emissions with an equivalence ratio of ϕ = 0.6 to ϕ = 1.2 in lean and rich mixtures were measured by the Kane 251 gas analyzer. Figure 6 illustrates the CO and NO emission levels (ppm) in double-layer PMB with varying equivalence ratios. CO emissions for double-layer PMB is less than 225 ppm for all equivalence ratios which are acceptable limits except at ϕ = 0.8 and 0.9, where the values are 986 and 1052 ppm, respectively. This rapid increase in CO emissions is a result of the quenching effect on the burner surface and the short residence time of product gases in the combustion chamber [17]. CO emissions increase with increasing quenching on the burner surface. As shown in Table 2, where the *Pe* number for alumina foam is less than 65 in the reaction zone, the quenching effect occurs at both ϕ = 0.8 and 0.9. This effect occurs because the flow of the mixture was not uniformly supplied due to the pore size of the porous media and the inlet flow velocity being incompatible at both equivalence ratios. Additionally, CO emissions will increase as a consequence of the insufficient conversion of CO to CO_2_ due to the short oxidation time. As a result, the CO emissions increase significantly cause by incomplete combustion. The formation of CO emission is also negatively affected by the flame velocity. On the other hand, a greater flame velocity suppressed the residence time, which would significantly reduce the emission of CO. The flame velocity does not comparatively differ but the indicative wall temperature of the burner (Figure 5) shows a slight increment at ϕ = 0.8 and 0.9. The increment in the wall temperature has indirectly affected the formation of CO by enriching the alumina foam and unintendedly prolonged the residence time. It is also evidenced that the CO formation shows a steep decrease when the value of equivalence ratio was further increase toward the rich mixture. The effect of the residence time is no longer perceived as significant, and the interconnectivity among the beads has resulted in the overall reduction of CO at rich mixtures. 

As shown in Figure 6, the NO emission is relatively constant between the equivalence ratios of ϕ = 0.6 and ϕ = 0.9. The values recorded were 1 ppm to 6 ppm and these values began to increase significantly from ϕ = 0.9 (4 ppm) to ϕ = 1.1 (20 ppm), before a subtle drop at ϕ = 1.2 (17 ppm). The NO formations in the current work are due to the effect of fuel/air mixtures residence time and flame temperature. In the experimental range of ϕ < 1.0, the NO emission is very low and the PMB is operating optimally in the lean region of combustion. The flame velocity, *S_L_* in the lean region is noticeably higher compared with a rich mixture. Thus, the NO concentration is effectively being carried downstream of the burner instead of being confined inside the burner. A reduction in the flame velocity, *S_L_*, also resulted in an increase in the residence time and a decrease in the flame temperature. It occurs when the flame velocity is thought to move rather too slowly, resulting in a prolonged oxidation time of the mixture to attain complete combustion in the burner. This is attributed to the difference in the porosity between an alumina and porcelain foam in a double-layer PMB that has accelerated the NO emissions production [17,44]. Therefore, the residence time of the reaction mixtures was considerably increased, resulting in higher NO emissions.

### 3.4. Entropy Generation Rate, Energy Loss and Exergy Destroyed in Different Equivalence Ratio

Figure 7 shows the influence of the equivalence ratio on the total entropy generation, S_gen_, in a double-layer PMB premixed butane-air combustion. The overall entropy generation yielded the lowest value at an equivalence ratio of ϕ = 1.0 (stoichiometric mixture) as 0.0048 W/K. A uniform reduction in the entropy generation rate owing to the heat transfer conduction can be obtained by increasing the equivalence ratio from ϕ = 0.6 to a stoichiometric mixture of ϕ = 1.0. Thereafter, the entropy generation rate starts to increase significantly in rich combustion, from ϕ = 1.1 to ϕ = 1.2.

This result shows that an increase in the mass flow rate of air has no significant impact on the entropy generation (lean combustion), whereas an increase in the mass flow rate of butane has a greater influence on the entropy generation (rich combustion), as shown in Table 2. The rise in the surface flame temperature was caused by an increase in the mass fraction of the fuel, which implies that more chemical energy is released. As a consequence, the entropy generation rate increased from ϕ = 1.0 to 1.2, indicating that the butane fuel has provided greater heating value in the rich combustion region. Furthermore, when the temperature gradient moves from the upstream to downstream of the burner, the fuel with a greater equivalence ratio (rich mixtures) was easily ignited by flame combustion [29]. As a result, an increase in the butane fuel flow rate in the premixed combustion contributed toward an increase in the entropy generation rate, and it has affected the surface flame temperature. This result is comparable to that of Jiang et al. [30], who claimed that increase flow velocity increases thermal conduction, which caused the entropy generation rates. 

Figure 8 shows that a change in the energy loss, E_loss_ (energy wastes during the combustion) with variations in the equivalence ratio for a double-layer PMB. It is noticed that the energy loss climbed up to the maximum at ϕ = 0.8, and it dropped to its lowest value at ϕ = 1.0. Then, the energy loss started to climb up again at ϕ = 1.1 and 1.2. From the results obtained, the highest energy loss at ϕ = 0.8 is 117.40 W which is approximately 19.70% of the lowest energy loss in the lean combustion and only 1.88% in the rich combustion (at ϕ = 1.2). A possible reason is the effect of the radiative heat transfer from the burner wall, which acts to augment the heat loss from the combustion (the purple contour color around the burner housing). It can be seen from Figure 7 that at ϕ = 1.0, the entropy generation is at its lowest value, which implied that less energy is wasted from the premixed combustion. This also partially proved that at ϕ = 0.8 and 0.9, the porous wall temperature for both alumina and porcelain foam in the PMB becomes higher. As a result, the effects from the equivalence ratio and a double-layer PMB on energy loss, E_loss_, have significantly impacted the butane-air premixed combustion.

Figure 9 shows that the exergy destroyed, E_des_ with the variations in the equivalence ratio for a double-layer PMB. The exergy destroyed, which is attributed to the entropy generation, must be minimize in order to obtain better and improved energy performance in the PMB. Looking at Equation (11), the exergy destroyed obtained indicated that an increase in the equivalence ratio has resulted in the exergy destroyed reduces to the minimum at ϕ = 1.0 from ϕ = 0.6 in the lean mixtures and increases to the maximum at ϕ = 1.2 in the rich mixtures.

The flame temperature distribution at ϕ = 1.0 revealed that the flame stability is attained, with mild fluctuations in the flame temperature based on the four temperature recordings in Figure 4. Furthermore, as previously stated, the exergy destroyed increases as the butane fuel flow rate increases, which is affected on the surface flame temperature to increase in premixed combustion.

### 3.5. Thermal Efficiency and Exergy Efficiency

The performance of the PMB depends on the fraction of the thermal and exergy efficiency. The effect of the equivalence ratio from lean to rich mixtures on the double-layer PMB can be explained by considering Figure 10. From Equations (7) and (12), it can be observed that the thermal efficiency, η_Thermal_, and exergy efficiency, ƞ_Exergy_, reduced from ϕ = 0.6 to 0.8 but these efficiencies started to increase when the premixed butane-air equivalence ratios start to rise from 0.9 to 1.2. The highest thermal and exergy efficiency achieved at ϕ = 1.2 with 59.30% and 58.47%, respectively. The lowest thermal and exergy efficiency was found when ϕ = 0.8, with 42.63% and 41.87%, respectively. The result shows that the average efficiency for both of them is approximately 50%. The lowest thermal and exergy efficiency, at ϕ = 0.8 indicates that the largest energy loss during combustion led to the reduction of PMB performance, implying that the radiation heat loss occurs which surrounds the burner housing during combustion, as can be seen in Figure 3. It was also discovered that in porous media combustion, the larger energy loss inside combustion has resulted in the lowest performance efficiency [28]. The same results obtained by Yu et al. [17] claimed that thermal efficiency increases due to a reduction in energy loss at the exhaust outlet between ϕ = 0.80 and 0.95.

An increase in the efficiency as the equivalence ratio rises from ϕ = 0.9 to 1.2 depends on the surface flame combustion, which begins to deform on top of the burner surface at ϕ = 0.9. This is due to the fact that the container is placed on top of the PMB, which was where the water boiling test was conducted. It means that the flame temperature had begun to rise, and the flame location had shifted away from the porous surface. A reduction in the radiation heat transfer from the wall of the porous media burner as well as an increase in the equivalence ratio inside the premixed combustion are the primary contributors to an increase in the PMB performance efficiency. It can be inferred that energy loss, E_loss_, and the surface flame temperature of butane-air mixture combustion have a significant impact on the thermal efficiency of the PMB.

Figure 11 shows that the thermal images of a free flame and a porous media burner (PMB) at several values of equivalence ratios, while Table 4 compares the free flame and porous media combustion under both lean and rich mixtures in butane-air premixed combustion. As the equivalence ratio increase from lean to rich in PMB, each of them exhibited different thermal and exergy efficiency. It can be observed that there are several applications that are ideal for free flame and employing porous media burner in combustion systems.

## 4. Conclusions

In this study, the analysis of the premixed butane-air combustion in a 23 mm double-layer PMB by usability of entropy generation, thermal and exergy efficiency were carried out. The performance efficiency of porous medium burners with varying equivalence ratios was compared and analyzed using a simple governing equation to provided useful information on PMB. The main conclusions are as follows:The effects of the equivalence ratio and double-layer porous media from lean to rich mixtures has significant impact on the thermal and exergy efficiency, and therefore the flame stability and temperature change as equivalence ratio increases. The results demonstrated that the submerged flame occurred in a lean mixture inside a double-layer PMB with a reaction zone, resulting in a red heat wall burner. The maximum flame temperature ranged between 495.69 °C and 820.95 °C with the variations in the equivalence ratio, while porous wall temperature was found to be almost constant between 287.08 °C and 436.16 °C. The maximum overall thermal and exergy efficiencies are 59.30% and 58.47% at rich combustion. Moreover, the PMB operates at the optimal combustion at ϕ = 1.0, and produces the lowest entropy generation rate, energy loss, and exergy destroyed. Carbon monoxide, CO emission at ϕ = 0.8 and 0.9 rise significantly which indicated an incomplete combustion which cannot be regarded as an acceptable limit for human health because it could be seriously hazardous. 

The outcome of these results suggest that the modification of alumina foam thickness (reaction zone) coupled with the corresponding changes in the Peclet number, *Pe*, adversely affect the CO emission. This work also demonstrates the benefits of using porous media materials and a double-layer configuration to improve the performance of micro-burners in terms of thermodynamic irreversibility for butane-air premixed combustion.

## Figures and Tables

**Figure 1 entropy-23-01663-f001:**
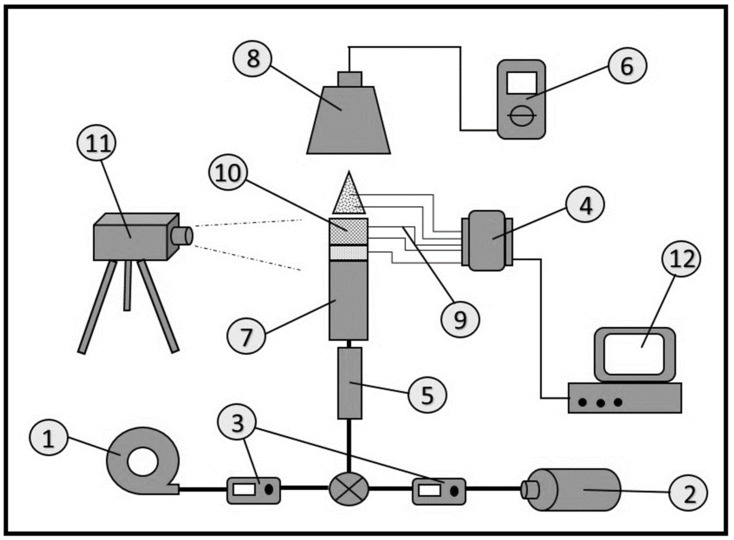
Schematic diagram of the PMB set-up. (1) Air pump, (2) Butane cartridge, (3) Flow meter controllers, (4) Data acquisition system (DAQ), (5) Mixing unit, (6) Gas analyzer, (7) Burner housing, (8) Smokestack collector, (9) K-type thermocouples, (10) Porous media, (11) Thermal imager, and (12) Personal computer.

**Figure 2 entropy-23-01663-f002:**
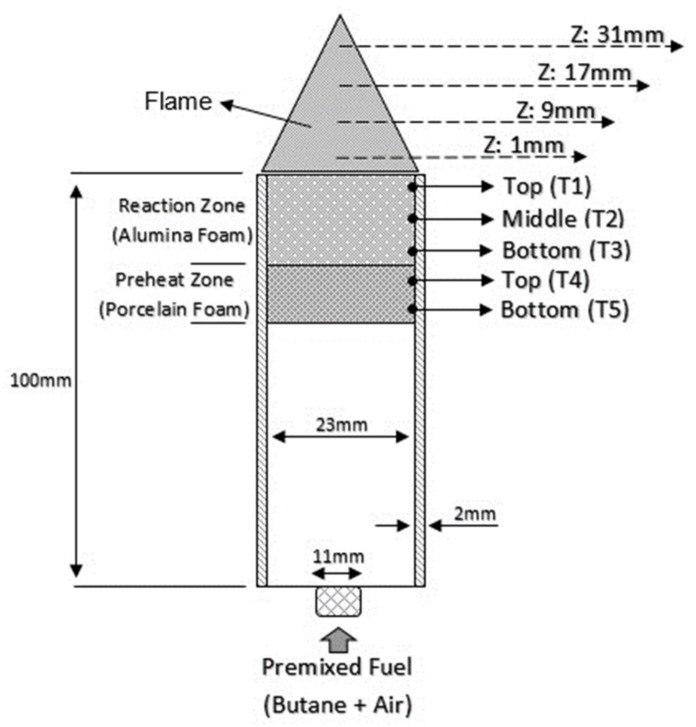
Schematic diagram of the double-layer PMB configuration with thermocouples placement.

**Figure 3 entropy-23-01663-f003:**
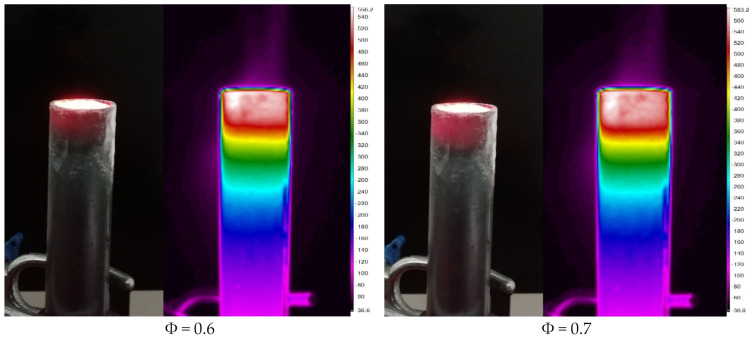
Photographs (**left**) and thermal (**right**) images of the double-layer PMB configuration with varies equivalence ratios.

**Figure 4 entropy-23-01663-f004:**
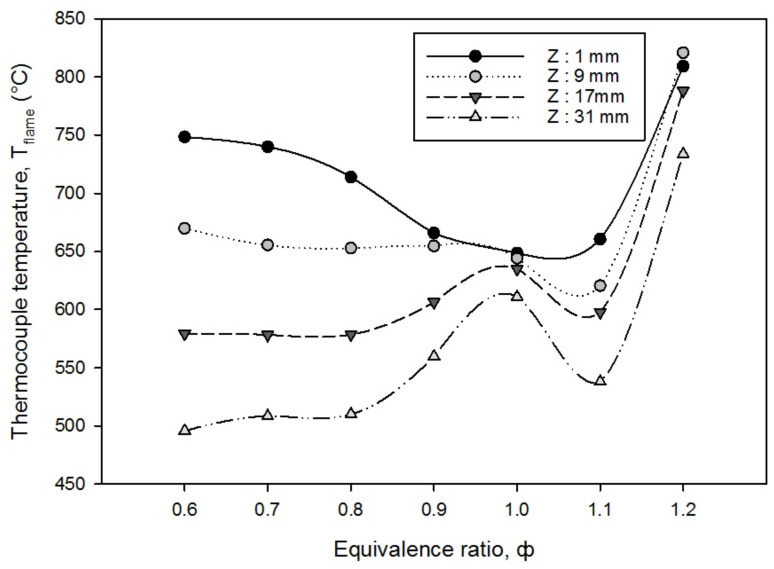
The flame temperature of the top surface PMB at different thermocouples height, Z with a variation in the equivalence ratios.

**Figure 5 entropy-23-01663-f005:**
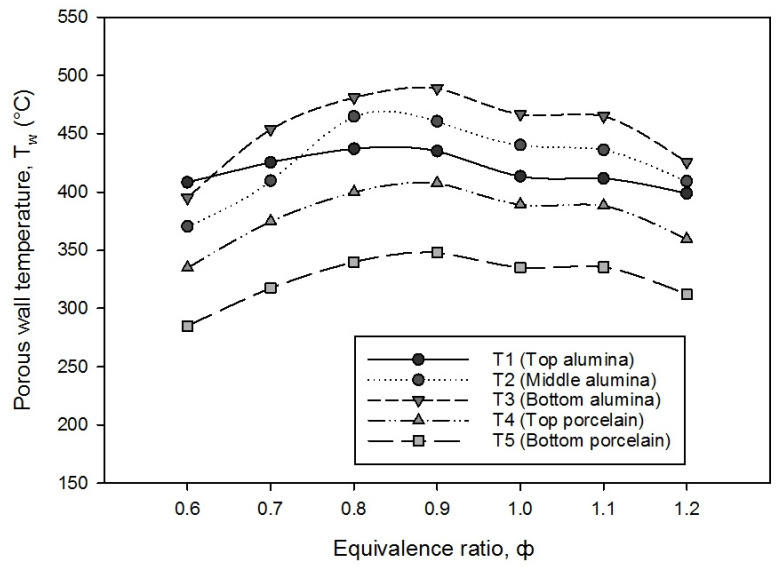
The porous wall temperature on different porous media foam with varies equivalence ratios.

**Figure 6 entropy-23-01663-f006:**
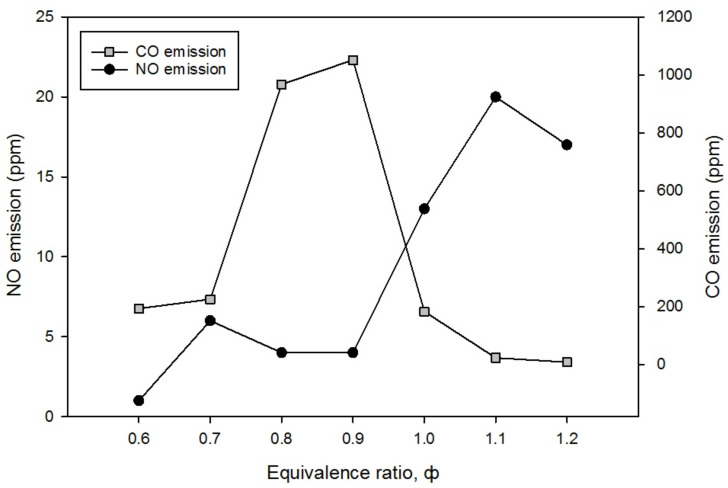
The CO and NO emission levels (ppm) with varies equivalence ratios.

**Figure 7 entropy-23-01663-f007:**
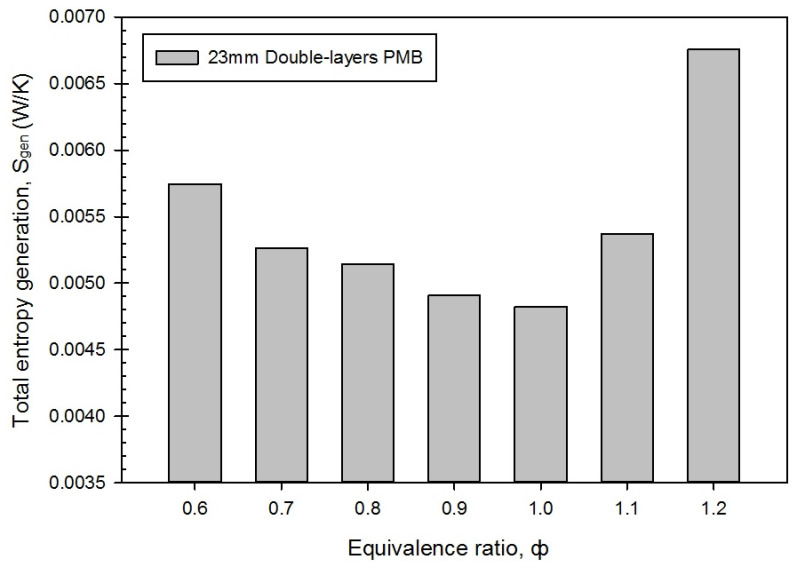
Total entropy generation, S_gen_ with varies equivalence ratios.

**Figure 8 entropy-23-01663-f008:**
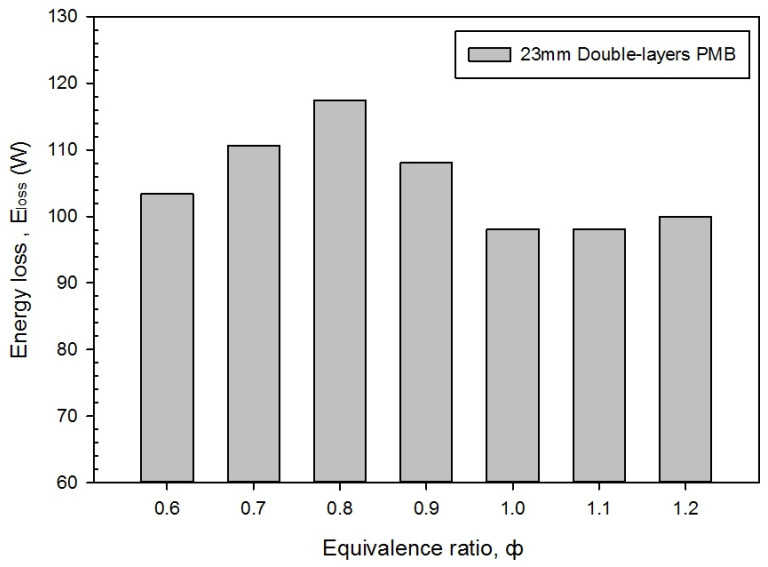
Energy loss, E_loss_ with varies equivalence ratio.

**Figure 9 entropy-23-01663-f009:**
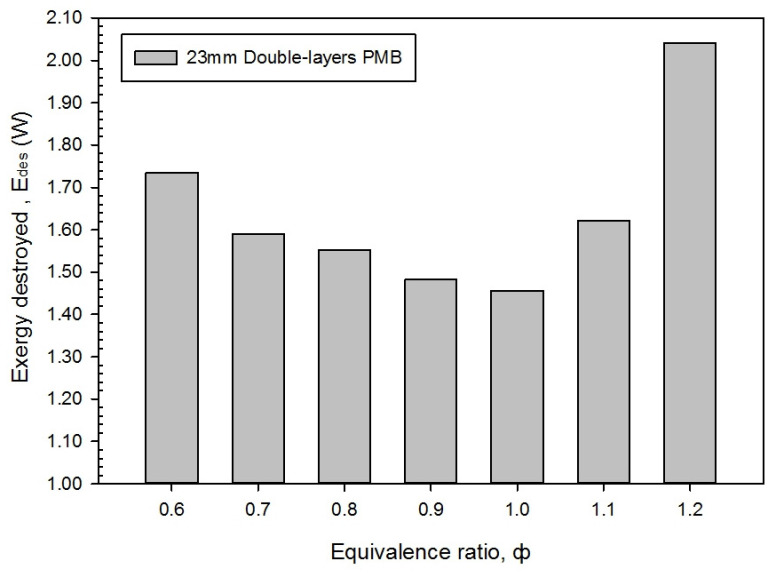
Exergy destroyed, E_des_ with varies equivalence ratios.

**Figure 10 entropy-23-01663-f010:**
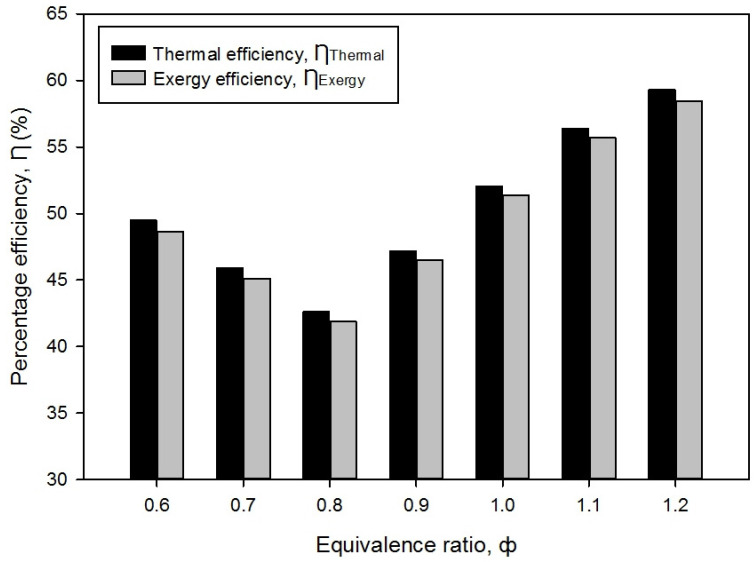
Thermal efficiency, η_Thermal_ and exergy efficiency, η_Exergy_ with varies equivalence ratios.

**Figure 11 entropy-23-01663-f011:**
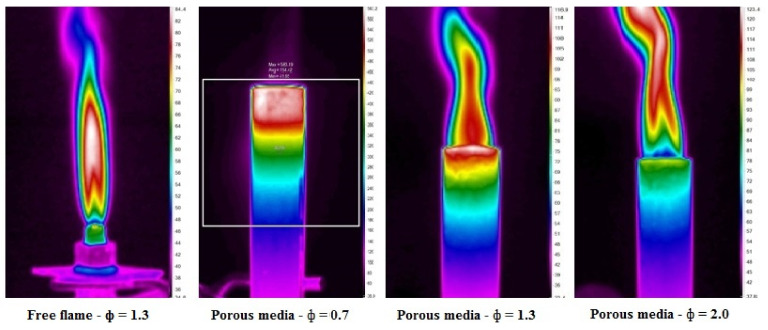
Thermal images of free flame and porous media burner (PMB) with varies equivalence ratio.

**Table 1 entropy-23-01663-t001:** Recent studies of effect on lean and rich equivalence ratio on PMB systems.

Authors	Equivalence Ratio	Fuel/Oxidizer	Comments
Peng et al. [21]	0.9 < ϕ < 1.1	H_2_/Air	Inserting porous media or increasing the outer wall thickness can improve heat transfer in a micro-combustor and affecting flame stability.
Liu et al. [22]	0.5 < ϕ < 0.9	CH_4_/Air	An increase in wall thermal conductivity, both the upper and lower limits for the standing wave regime rise.
Pan et al. [23]	ϕ = 0.6, 0.8, 1.0	H_2_/O_2_	Effects of equivalence ratio, mixture flow rate, and porosity can increase the combustion efficiency.
Mansir et al. [24]	0.4 < ϕ < 1.0	CH_4_-CO_2_/O_2_	The development of an enhanced mixing zone adjacent to the porous plate improved flame anchoring and stability.
Li et al. [25]	ϕ = 1.0	H_2_/Air	Heat recirculation through the combustor wall decreases as combustor dimension decreases.
Qian et al. [26]	ϕ = 0.6, 0.8, 1.0, 1.2	H_2_/Air	The porous media combustor with the bluff-body performs better in terms of system efficiency and blowout limit.
Qu et al. [27]	0.25 < ϕ < 0.35 0.55 < ϕ < 0.70	CH_4_/Air C_3_H_8_/Air H_2_/Air	The flame stability limits of methane, propane, and hydrogen increase as the equivalence ratio arise.

**Table 2 entropy-23-01663-t002:** The summary of the parameters studied in the PMB with variation in the equivalence ratio.

Equivalence Ratio, ϕ	0.6	0.7	0.8	0.9	1.0	1.1	1.2
Butane fuel (liters/min)	0.10					0.11	0.12
Air (liters/min)	5.16	4.42	3.88	3.44	3.10		
Flame velocity, *S_L_* (m/s)	0.2086	0.1805	0.1564	0.1404	0.1284	0.1288	0.1292
Reynolds number, *Re*	684	592	513	461	421	423	424
Peclet Number, *Pe*							
Alumina foam	78.93	68.30	59.18	53.12	48.58	48.74	48.89
Porcelain foam	23.99	20.76	17.99	16.15	14.77	14.82	14.86

**Table 3 entropy-23-01663-t003:** Maximum uncertainty analysis of flame temperature, T_flame_ and porous wall temperature, T_w_ with varies thermocouples placement.

Variables	Mean, $\bar{x}$ (°C)	Standard Deviation, $\sigma_{x}$	Standard Error, $\sigma_{\bar{x}}$	Uncertainty, U_n_ (%)
**Thermocouple height, Z**	Flame temperature, T_flame_			
**1 mm**	809.29	1.153	0.665	0.142
**9 mm**	644.12	3.747	2.163	0.582
**17 mm**	634.87	3.312	2.771	0.756
**31 mm**	610.82	22.154	12.790	3.627
**Porous wall Temperature, T_w_**				
**Alumina foam**	370.53	8.391	4.845	2.265
**Porcelain foam**	334.93	5.444	4.086	2.113

**Table 4 entropy-23-01663-t004:** Comparison parametric studies between free flame and porous media combustion (PMB).

Parametric Studies	Free Flame [47]	PMB − ϕ = 0.7	PMB − ϕ = 1.3 [47]	PMB − ϕ = 2.0 [47]
Combustion mode	Rich	Lean	Rich	Rich
Flame type	Diffusion flame	Submerged flame	Surface flame	Surface flame
Flame color	Light blue	Glowing red heat	Blue	Orange
Energy produce, Q_actual_ (W)	152.57	93.95	191.12	344.95
Thermal efficiency (%)	57.40	45.91	71.80	84.30
Exergy efficiency (%)	56.48	45.13	70.95	83.47
Max. flame temperature (°C)	891.38	740.05	924.82	825.54
CO emission (ppm)	9	225	11	16
NO emission (ppm)	7	6	5	11

## Data Availability

Not applicable.

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
