# Peer review of "Double-Layer Micro Porous Media Burner from Lean to Rich Fuel Mixture: Analysis of Entropy Generation and Exergy Efficiency"

_entropy, 2021, doi:10.3390/e23121663_

Round 1

Reviewer 1 Report

This study appears to be sound, and it is rather well described, even though it is rather bloated and could be shortened quite a bit. For instance, it is not necessary to mention the second law of thermodynamics over and over again. Or, in section 2.3 it is mentioned that a lab thermometer was used. As opposed to what: a cooking thermometer, or a lab barometer? Also, im my eyes Figure 2 is entirely useless.

a few more hints: 

page 1 line 44: you mean "increase" instead of "increased".

the same: lines 67, 93.

page 7 line 197: you mean "values were" instead of "values was".

page 10 line 267: you mean "increase" instead of "increased".

the same: lines 280, 287, TWICE line 296, line 298

the same: lines 357, 374, 375, 376, 380, 381, 400, 418

the same: TWICE line 422

line 348: you mean "decrease" instead of "decreased".

the same: line 357

Author Response

The authors would like to thank you and all the reviewers for their detailed and constructive comments to further strengthen this manuscript. We are extremely grateful to all reviewers for the effort and precious time put into the review of this manuscript. Each comment has been carefully considered and responded. The corresponding changes made in the revised manuscript are summarized below. The comments from all reviewers are in normal font and our responses are italicized.

Comments and Suggestions for Authors

This study appears to be sound, and it is rather well described, even though it is rather bloated and could be shortened quite a bit. For instance, it is not necessary to mention the second law of thermodynamics over and over again.

Response:

We fully agree with the reviewer that the contents should be shortened to improve the clarity for the general audience of Entropy. Your suggestion is greatly helpful, and we have removed the mentioned repeating words in the manuscript.

Or, in section 2.3 it is mentioned that a lab thermometer was used. As opposed to what: a cooking thermometer, or a lab barometer?

Response:

Thanks for the comments. We have rewritten the sentence for better reflecting the actual meaning of the equipment.

Also, im my eyes Figure 2 is entirely useless.

Response:

Thank you. We have removed Figure 2 from the manuscript

a few more hints: 

page 1 line 44: you mean "increase" instead of "increased".

the same: lines 67, 93.

page 7 line 197: you mean "values were" instead of "values was".

page 10 line 267: you mean "increase" instead of "increased".

the same: lines 280, 287, TWICE line 296, line 298

the same: lines 357, 374, 375, 376, 380, 381, 400, 418

the same: TWICE line 422

line 348: you mean "decrease" instead of "decreased".

the same: line 357

Response:

Thank you for highlighting the grammar mistakes in the manuscript. We have made necessary amendment according to the recommendations by the reviewer.

Reviewer 2 Report

The main objective and the research direction followed by the researchers in this research paper is to investigate the entropy generation, thermal and exergy efficiency in double-layered porous media burners (PMBs) that utilizes premixed butane-air combustion with a detailed equivalence ratio.  This research study provides useful information on the design and operation of PMBs, as well as substantial knowledge of the combustion 115 phenomena in a double-layer PMB.   The topic is novel as the experimental investigation is carried out and real-time results are presented. Moreover, this work is novel in the sense that this can also demonstrate the benefits of using porous media materials and a double-layer configuration to improve the performance of micro-burners in terms of  thermodynamic irreversibility for butane-air premixed combustion.   Paper is well written with a proper flow and the conclusion are perfectly aligned with the methodology, aims and research objectives.

Author Response

The authors would like to thank you and all the reviewers for their detailed and constructive comments to further strengthen this manuscript. We are extremely grateful to all reviewers for the effort and precious time put into the review of this manuscript. Each comment has been carefully considered and responded. The corresponding changes made in the revised manuscript are summarized below. The comments from all reviewers are in normal font and our responses are italicized.

Comments and Suggestions for Authors

The main objective and the research direction followed by the researchers in this research paper is to investigate the entropy generation, thermal and exergy efficiency in double-layered porous media burners (PMBs) that utilizes premixed butane-air combustion with a detailed equivalence ratio.  This research study provides useful information on the design and operation of PMBs, as well as substantial knowledge of the combustion 115 phenomena in a double-layer PMB.   The topic is novel as the experimental investigation is carried out and real-time results are presented. Moreover, this work is novel in the sense that this can also demonstrate the benefits of using porous media materials and a double-layer configuration to improve the performance of micro-burners in terms of thermodynamic irreversibility for butane-air premixed combustion.   Paper is well written with a proper flow and the conclusion are perfectly aligned with the methodology, aims and research objectives.

Response:

We are extremely grateful to the reviewer for the positive comments. Thank you for the commendable made and your comments have fuelled our motivation to achieve greater heights in this research.

Reviewer 3 Report

Comment to the paper: “Double-layer Micro Porous Media Burner from Lean to Rich Fuel Mixture: Analysis of Entropy Generation and Exergy Efficiency” by Ismail et al.

General comment:

The manuscript deals with investigations on double-layer micro porous media burner by using butane-air combustion in order to study the entropy generation, thermal and exergy efficiency. The effect of the equivalent ratio was tested in terms of flame characteristics, temperature variation, emissions, energy loss, and exergy destroyed.

The manuscript is very interesting and suitable to be published in this journal; however, some points should be addressed before publication.

Some minor language mistakes are present that should anyway be corrected.

Please, use the same font for the template.

  1. Results and discussion

Please, add standard deviation.

Please, compare the proposed approach with literature data.

Please, compare results with literature.

Author Response

The authors would like to thank you and all the reviewers for their detailed and constructive comments to further strengthen this manuscript. We are extremely grateful to all reviewers for the effort and precious time put into the review of this manuscript. Each comment has been carefully considered and responded. The corresponding changes made in the revised manuscript are summarized below. The comments from all reviewers are in normal font and our responses are italicized.

Comments and Suggestions for Authors

Comment to the paper: “Double-layer Micro Porous Media Burner from Lean to Rich Fuel Mixture: Analysis of Entropy Generation and Exergy Efficiency” by Ismail et al.

General comment:

The manuscript deals with investigations on double-layer micro porous media burner by using butane-air combustion in order to study the entropy generation, thermal and exergy efficiency. The effect of the equivalent ratio was tested in terms of flame characteristics, temperature variation, emissions, energy loss, and exergy destroyed.

The manuscript is very interesting and suitable to be published in this journal; however, some points should be addressed before publication.

Some minor language mistakes are present that should anyway be corrected.

Please, use the same font for the template.

  1. Results and discussion

Please, add standard deviation.

Response:

We have added the calculation using the standard deviation formula as shown below. Thanks for the comment.

For all experimental data, the mean value  and the standard deviation  are expressed as follows:

Uncertainty is defined as:

Please, compare the proposed approach with literature data.

Response:

The reviewer raises a valid comment and proposed a scientific comparison between our approach and the published literature. Our methodology has been strengthened to align with several published works:

  1. Mishra et al. (2018)
  2. Su et al. (2016)
  3. Sharma et al. (2016)
  4. Janvekar et al. (2017)

Please, compare results with literature.

Response:

Good suggestion is raised by the reviewer. We also note that part of our discussion in the manuscript was not properly based on concrete findings by other researchers. We have therefore rewritten some of the sentences to enhance the readers’ understanding and interpretation of our results.

  1. This result is comparable to that of Jiang et al. [30], who claimed that increase flow velocity increases thermal conduction, which caused the entropy generation rates.
  2. This is known as flat flame behavior same reported by Giurcan et al. [16], in which the double-layer PMB acts as a flat flame burner by placing the porous media as a diffuser on the top of the burner rim, creating a constant profile of the air-fuel mixtures flow rate.
  3. The same results obtained by Yu et al. [17] claimed that thermal efficiency will rise due to a reduction in energy loss at the exhaust outlet between Ñ„ = 0.80 and 0.95.
